# Dairy-Derived Emulsifiers in Infant Formula Show Marginal Effects on the Plasma Lipid Profile and Brain Structure in Preterm Piglets Relative to Soy Lecithin

**DOI:** 10.3390/nu13030718

**Published:** 2021-02-24

**Authors:** Nicole L. Henriksen, Karoline Aasmul-Olsen, Ramakrishnan Venkatasubramanian, Mikkel K. E. Nygaard, Richard R. Sprenger, Anne B. Heckmann, Marie S. Ostenfeld, Christer S. Ejsing, Simon F. Eskildsen, Anette Müllertz, Per T. Sangild, Stine B. Bering, Thomas Thymann

**Affiliations:** 1Comparative Pediatrics and Nutrition, Department of Veterinary and Animal Sciences, University of Copenhagen, Dyrlægevej 68, 1870 Frederiksberg C, Denmark; nlh@sund.ku.dk (N.L.H.); karolineaas@sund.ku.dk (K.A.-O.); pts@sund.ku.dk (P.T.S.); sbb@sund.ku.dk (S.B.B.); 2Physiological Pharmaceutics, Department of Pharmacy, University of Copenhagen, Universitetsparken 2, 2100 Copenhagen Ø, Denmark; venkatasubramanian.ramakrishnan@sund.ku.dk; 3Center of Functionally Integrative Neuroscience, Department of Clinical Medicine, Aarhus University, Universitetsbyen 3, 8000 Aarhus C, Denmark; mken@cfin.au.dk (M.K.E.N.); seskildsen@cfin.au.dk (S.F.E.); 4VILLUM Center for Bioanalytical Sciences, Department of Biochemistry and Molecular Biology, University of Southern Denmark, Campusvej 55, 5230 Odense M, Denmark; richards@bmb.sdu.dk (R.R.S.); cse@bmb.sdu.dk (C.S.E.); 5Arla Foods Ingredients, Sønderhøj 10-12, 8260 Viby J, Denmark; anhec@arlafoods.com (A.B.H.); mstos@arlafoods.com (M.S.O.); 6Bioneer:FARMA, Department of Pharmacy, University of Copenhagen, Universitetsparken 2, 2100 Copenhagen Ø, Denmark; anette.mullertz@sund.ku.dk

**Keywords:** bovine whey phospholipids, polar lipids, neurodevelopment, brain, preterm, piglet, infant formula, emulsification, milk fat globule membrane, extracellular vesicles

## Abstract

Breastfed infants have higher intestinal lipid absorption and neurodevelopmental outcomes compared to formula-fed infants, which may relate to a different surface layer structure of fat globules in infant formula. This study investigated if dairy-derived emulsifiers increased lipid absorption and neurodevelopment relative to soy lecithin in newborn preterm piglets. Piglets received a formula diet containing soy lecithin (SL) or whey protein concentrate enriched in extracellular vesicles (WPC-A-EV) or phospholipids (WPC-PL) for 19 days. Both WPC-A-EV and WPC-PL emulsions, but not the intact diets, increased in vitro lipolysis compared to SL. The main differences of plasma lipidomics analysis were increased levels of some sphingolipids, and lipid molecules with odd-chain (17:1, 19:1, 19:3) as well as mono- and polyunsaturated fatty acyl chains (16:1, 20:1, 20:3) in the WPC-A-EV and WPC-PL groups and increased 18:2 fatty acyls in the SL group. Indirect monitoring of intestinal triacylglycerol absorption showed no differences between groups. Diffusor tensor imaging measurements of mean diffusivity in the hippocampus were lower for WPC-A-EV and WPC-PL groups compared to SL indicating improved hippocampal maturation. No differences in hippocampal lipid composition or short-term memory were observed between groups. In conclusion, emulsification of fat globules in infant formula with dairy-derived emulsifiers altered the plasma lipid profile and hippocampal tissue diffusivity but had limited effects on other absorptive and learning abilities relative to SL in preterm piglets.

## 1. Introduction

Preterm infants have a higher risk of developing long-term motor, cognitive and behavioural impairments compared to term-born infants [1,2,3]. Rapid brain growth in late gestation and early postnatal life is associated with increased vulnerability to nutritional deficiencies suggesting that nutrition may affect neurodevelopment [4]. Meta-analyses of clinical studies have demonstrated neurodevelopmental advantages to human breast milk relative to infant formula [5,6]. Due to their quantity and bioactive functions in the brain, lipid molecules, especially dietary-derived polyunsaturated fatty acids (PUFAs), have been investigated as modulators of cognitive development in infants [7,8]. Studies of the fatty acid composition of the cerebral cortex and cerebellum of infants that have died unexpectedly have shown higher concentrations of PUFAs in breastfed relative to formula-fed infants [9,10]. However, meta-analyses of randomized control trials evaluating the effect of fortification of infant formula with PUFAs have shown inconclusive results for cognitive function [11,12]. This could in part be explained by reduced intestinal triacyl-glyceride (TG) hydrolysis and absorption in formula-fed infants compared to breastfed infants [13,14]. 

In breast milk, fat globules are emulsified by a naturally occurring phospholipid membrane structure known as the milk fat globule membrane (MFGM). The MFGM is a heterogeneous triple-layered structure composed of polar lipids, cholesterol, proteins and glycoproteins derived from the endoplasmic reticulum and apical plasma membrane of the mammary gland epithelial cells [15]. Extracellular vesicles (EVs), comprising exosomes and micro-vesicles, represent another polar-lipid-rich source in milk and have gained interest for their potential roles in intercellular communication and human health and disease [16]. These bi-layered vesicles carry a variety of biomolecules including nucleic acids, proteins and lipids and are compositionally similar to the membrane of origin, although they generally seem to be enriched in cholesterol, sphingolipids, phosphatidylserine and proteins such as tetraspanins [17,18]. Studies suggest that micro-vesicles bud off directly from the plasma membrane, whereas exosomes are formed by inward budding of endosomes and external release [18]. This creates compositional differences between MFGM and subtypes of EVs [19]. In infant formula, soy lecithin (SL) in combination with proteins is often used as the emulsifier of TG droplets [20]. SL is extracted from soybean oil and mainly consists of the phospholipid phophatidyl-choline and has a distinct composition of fatty acyl chains compared to dairy-based emulsifiers [21]. Milk-derived semi-purified polar lipid fractions (80.2% polar lipids) with very little protein have been shown to improve gastric lipase activity in vitro relative to SL [20]. This highlights the potential for reducing differences in lipid hydrolysis and absorption between formula- and breastfed infants by modifying the surface layer of dietary fat particles in infant formula [20]. Formula-based diets enriched in polar lipids such as glycerophospholipids, sphingolipids and gangliosides have in some studies been shown to improve brain development and cognition in both infants [22,23,24,25] and animal models [26,27,28]. However, others were unable to demonstrate these effects [29]. Similar research investigating the neurodevelopmental effects of supplementing infant formula with EVs is limited. 

In this study, we used a preterm piglet model. The anatomy and growth trajectories of the piglet brain [30,31], early life nutritional requirements of piglets [32] and neurodevelopmental delays observed in preterm piglets [33,34] suggest that the preterm piglet is a good model for studying the effects of nutrition on neurodevelopment in late preterm human infants [33]. In the present study, we investigated if dairy-derived emulsifiers (whey protein concentrate from acid whey enriched in extracellular vesicles, WPC-A-EV, and whey protein concentrate enriched in phospholipids, WPC-PL) affects neurodevelopment in preterm piglets relative to SL when administered in a formula diet from birth to 19 days of age. Specifically, we tested the hypothesis that neurodevelopment is improved by increasing intestinal lipid uptake through modification of the surface of dietary fat particles. We found that WPC-A-EV and WPC-PL emulsifiers altered plasma lipid composition and lowered hippocampal tissue diffusivity indicating improved hippocampal maturation, but did not affect short-term memory in preterm piglets relative to the SL emulsifier. 

## 2. Materials and Methods

### 2.1. Study Design

Seventy-four piglets (Danish landrace, Duroc and Yorkshire mix; 33 female and 41 male) from three litters were delivered preterm by caesarean section at approximately 90% of gestation (day 106 of expected term day 117). Within three hours of delivery, piglets were fitted with orogastric feeding tubes (06FR, E-Vet, Haderslev, Denmark) and arterial umbilical catheters (04FR, E-Vet) [35]. Piglets were stratified into three treatment groups according to birth weight and sex. Each group was randomly assigned to receive a milk formula diet containing either WPC-A-EV (*n* = 24), WPC-PL (*n* = 25) or SL (*n* = 25) as the emulsifier for 19 days. Treatment groups were blinded to study participants. During the study period, in vitro lipolysis, intestinal lipid absorption, home cage activity and basic motor skill acquisition were evaluated, and spatial T-maze and novel object recognition tests were conducted. Piglets were euthanized on day 19 by intracardiac injection of pentobarbital sodium (Euthanimal Vet, 400 mg/mL, Alfasan, Woerden, The Netherlands). Following necropsy, organs were weighed, and brain tissue and plasma samples were collected for ex vivo magnetic resonance imaging (MRI) and lipidomics analysis. 

### 2.2. Nutrition 

Diets consisted of whey protein (Lacprodan^®^ DI-9224), casein (Miprodan^®^ 40), lactose (Variolac^®^ 855) (all from Arla Food Ingredients, Viby J, Denmark) and vitamins/minerals (Phlexy-Vits, Nutricia, Amsterdam, The Netherlands). An emulsion containing vegetable oils (Akonino, AAK, Malmö, Sweden; Arasco and MEG-3, DSM, Mulgrave, BC, Canada) and one of three fat emulsifiers (SL (AAK), WPC-PL or WPC-A-EV (both from Arla Food Ingredients)) were added. Each emulsion was comprised of (*w*/*w*) 1% emulsifier, 10% vegetable oils (98.71% Akonino, 1% MEG-3, 0.29% ARASCO) and 89% water and was prepared as previously described [36]. The energy and macro nutrient composition of the base diet was adjusted to partially mimic sow milk [32]. Table 1 and Table 2 describe the gross composition of the diets and emulsifiers, respectively. 

Piglets were fed the diets as minimal enteral nutrition every 2–3 h alongside fat-free parenteral nutrition. The parenteral nutrition was based on Kabiven 2053 mL bags with added Vamin, Soluvit, Peditrace, Vitalipid and 2.780 mM glucose (all from Fresenius Kabi, Uppsala, Sweden). From day 1–7, piglets were fed decreasing amounts of parenteral nutrition (120–48 mL/kg/day) and increasing amounts of enteral nutrition (32–128 mL/kg/day). On day 8, the arterial umbilical catheters were removed, and piglets were weaned from parenteral nutrition and switched to total enteral nutrition (128–180 mL/kg/day) which continued until day 19. Piglets were trained to drink from feeding troughs from day 5 allowing the orogastric feeding tubes to be removed on day 11. Piglets had access to water ad libitum from day 5–19 and were supplemented with an oral electrolyte mixture (Revolyt, Gunnar Kjems, København, Denmark) and 2.780 mM glucose solution (Fresenius Kabi) twice daily from day 8–19. 

### 2.3. Animal Housing and Treatment

Piglets were housed individually in incubators from day 1–5, in larger boxes from day 5–11 and in larger cages from day 11–19. Incubators, boxes and cages were equipped with a heat source, soft pads, diapers and cloths/toys for enrichment. The clinical status of piglets was evaluated and scored twice daily on a scale from 1–4 (best to worst). Body weight was recorded daily, and temperature was measured regularly after birth until stable and thereafter upon indication. A feces score was given twice daily (0: no feces, 1: meconium or normal firm feces, 2: soft feces, 3: pasty feces, 4: diarrhea). Within the first 12 h of life, piglets were infused intra-arterially with sow-derived plasma at 25 mL/kg to provide passive immunization [35]. To prevent early onset diarrhea caused by bacterial pathogens, all piglets were treated prophylactically with a combination of oral antibiotics: gentamicin (Gentocin Vet, 4.36 mg/mL, 0.6 mL/kg, Scanvet, Fredensborg, Denmark), amoxicillin/clavulanic acid (Bioclavid, 500 + 125 mg/20 mL, 1 mL/kg, Sandoz International GmbH, Holzkirchen, Germany) and metronidazole (Flagyl, 40 mg/mL, 0.3 mL/kg, Sanofi A/S, Paris, France) twice daily for 3 days during the first 10 days. Piglets received additional antibiotics at a later stage if they had ongoing diarrhea. Piglets were euthanized if any of the following humane endpoints were reached: no pain relief, respiratory distress, severe ongoing diarrhea or symptoms of sepsis with no response to treatment. 

### 2.4. In Vitro Lipolysis and Serum Triacyl-Glyceride Measurements

The digestion of the pure emulsions and intact diets were evaluated using a two-step pediatric in vitro digestion model as previously described [37]. The compositions of the simulated gastric and intestinal media are presented in Appendix A. Briefly, the in vitro digestion was carried out in a thermostated vessel maintained at 37 °C. The pH during the gastric (0–50 min) and intestinal (51–140 min) phase was maintained at 6.4 and 6.5, respectively, using 0.2 M sodium hydroxide as a titrant. The simulated gastric digestion was initiated by adding 3.75 TBU/mL of recombinant human gastric lipase (rHGL) (Bioneer A/S, Hørsholm, Denmark) and 126 U/mL of porcine pepsin (Sigma-Aldrich, St. Louis, MO, USA) to 9.5 mL of emulsion/intact diet with 0.9 mL of concentrated simulated gastric medium. After 50 min of the gastric step, 5.85 mL of concentrated simulated intestinal media was added which simulated the transfer from gastric to intestinal compartment. The simulated intestinal digestion was initiated by adding 26.5 TBU/mL of porcine pancreatic extract (Sigma-Aldrich). To account for the untitrated non-esterified fatty acids during the gastric and intestinal step, the pH was raised to 9.0 at the end of the digestion [38].

As an indirect measure of intestinal lipid absorption, serum TG levels were determined on day 4. A 5-h enteral fasting blood sample was taken from the arterial umbilical catheter, and piglets were thereafter fed a normal feeding bolus of their respective diets (8 mL/kg). Blood was sampled at 30, 60 and 90 min after the feeding bolus. These time points were chosen based on previous studies in young pigs fed milk formula diets that have shown a peak in postprandial plasma TG levels at around 1 h [36,39]. Piglets continued to receive parenteral nutrition throughout. 

### 2.5. Magnetic Resonance Imaging

The right hemisphere of each brain was fixed in 10% neutral buffered formalin and placed in phosphate-buffered saline for 7–10 days prior to MRI. Diffusion tensor imaging (DTI) was obtained using a Stejskal-Tanner sequence (TR/TE = 5500/18 ms, average = 1, spatial resolution = 300 μm × 300 μm × 300 μm) on a 9.4T BioSpec 94/30 USR spectrometer (Bruker BioSpin, Ettlingen, Germany) equipped with a 1500 mT/m gradient system and a 40-mm inner diameter transmit-receive volume coil. Motion probing gradients were applied in 24 noncollinear directions with b = 3000 s/mm^2^ in addition to 4 b0. To cover the entire brain, two 3D imaging stacks were acquired, co-localized with ITK-snap [40] and stitched together with Python and Numpy. Reconstruction and DTI analysis (fractional anisotropy and mean diffusivity maps) of the stitched DTI data were performed in the Diffusion Toolkit software (version 0.6.4, TrackVis.org, Martinos Center for Biomedical Imaging, Massachusetts General Hospital, Boston, MS, USA). The previously reported MRI template [34] was used for defining eight regions of interest (nucleus accumbens, hippocampus, amygdala, caudate nucleus, lentiform nucleus, fornix, prefrontal cortex, internal capsule). For the current work, the template was extended to improve the registration from DTI to the 3D spoiled gradient-based template. The DTI images from the original 15 individual piglets used for the template were denoised [41], corrected for Gibbs ringing [42], bias field corrected [43], and diffusion metrics were estimated using weighted least squares [44]. The fractional anisotropy maps were transformed to the template using the spoiled gradient echo based transformations previously estimated [34] and averaged into a fractional anisotropy template. DTI b0 and fractional anisotropy images of the present study were nonlinearly registered to the extended template using a multimodality registration in ANTS [45]. The atlas labels of the template were then warped to DTI space and visually inspected before calculating the volume and mean of DTI parameters within each region of interest. 

### 2.6. Mass Spectrometry-Based Lipidomics

Plasma and hippocampus tissue samples were collected 1 h after piglets received a 15 mL/kg dietary bolus. Lipids were extracted from plasma and hippocampus tissue samples and analyzed by high-resolution shotgun lipidomics as previously described [46,47,48,49]. In total, 6–7 randomly chosen biological replicates per group were analyzed. 

### 2.7. Acquisition of Basic Motor Skills, Home Cage Activity and Short-Term Memory

The time it took to acquire basic motor skills (first stand, walk and drink from trough) was registered for each piglet. Home cage activity was recorded from day 2–5 using infrared surveillance cameras and analyzed as the proportion of active time in one-hour intervals by the PIGLwin software (Ellegaard Systems, Faaborg, Denmark). A novel object recognition test was conducted on day 9 and 10 and consisted of three habituation phases and one test phase. On day 9, piglets were habituated for 2 × 5 min in a 120 × 120 cm arena. On day 10, piglets were habituated for 5 min in the same arena containing two identical objects (white plastic containers 29 × 18 × 13 cm). Piglets were then returned to their cages for 5 min whilst one of the two objects in the arena was replaced at random by a novel object (blue kettlebell 21 × 17 × 10 cm). Piglets were placed back into the arena for 5 min during which the amount of time spent in contact (snout contact) with the novel and familiar object was recorded manually as a measure of explorative behaviour and short-term recognition memory [50]. The time spent investigating the novel object relative to both objects combined in the test phase (the recognition index) [50] as well as the latency to first touch of objects were determined. In between tests, the arena and objects were cleaned with soap and water, sprayed with ethanol and dried. Piglets were excluded from analyses if they were in contact with either object in the habituation or test phase for less than 2 s. From day 12–17, spatial working memory was assessed in a spatial T-maze test [51]. Prior to the test, piglets were habituated to the new surroundings and human handling. Piglets were placed in alternating starting positions and had to orientate themselves using visual cues placed outside the maze (posters with different patterns and colors) in order to localize a fixed positioned milk reward. Piglets had to complete 10 trials per day and the learning criteria was set at 80% correct choices per day. Tests were recorded using the software EthoVision XT10 (Noldus, Wageningen, The Netherlands). Piglets that displayed continuous off-task behaviour and no interest in the milk reward were excluded from the analyses. In both memory tests, piglets were tested in a randomized order. 

### 2.8. Statistics

Data analysis was performed using the statistical software R (version 3.6.1, R Foundation for Statistical Computing, Vienna, Austria), and graphical illustrations were done in GraphPad Prism (version 8.3.0, GraphPad Software, La Jolla CA, USA). Repeated measurements over time for continuous variables (NaOH, serum TGs, home cage activity, T-maze score and body weight) were analyzed using linear mixed effects models with a gaussian and exponential correlation for NaOH and body weight, respectively. Continuous novel object recognition test, MRI, lipidomics, organ weight and remaining in vitro data were analyzed using linear models. The incidence of diarrhea and proportion of piglets receiving additional antibiotics were evaluated by generalized linear models. Discrete time-to-event data (T-maze 80% learning criteria) were analyzed by a generalized linear mixed effects model, whilst continuous time-to-event data (basic motor skills) were analyzed using a cox proportional hazard model. A one-sample t-test was used to compare the novel object reference index to a null preference value of 0.5. Models of in vivo data included fixed effects such as treatment, birth weight, sex, litter and baseline TG levels and pig as a random effect. Models of in vitro data included repetition as a random effect. Model validation was conducted by testing normality and homoscedasticity of the residuals and fitted values, and if necessary data were transformed to meet these assumptions. Model validation of lipidomics data was conducted for a random subset of lipids representative of the full dataset. Non-parametric analysis was used when data could not be transformed. Lipidomics data were corrected for multiple comparisons using a false discovery rate correction. Additionally, post-hoc Tukey tests were used to correct for multiple comparisons between treatment groups. Data are presented as means with standard deviations. *p*-values below 0.05 were regarded as statistically significant.

## 3. Results

### 3.1. Clinical Outcomes, Growth and Organ Weights

Of the 74 piglets born, 14 piglets were euthanized/died during the 19-day study period due to reasons unrelated to the dietary treatment (necrotizing enterocolitis, catheter-related thrombosis, respiratory distress, seizures associated with hypoglycemia or unknown reasons) (Appendix A). All remaining piglets had clinical scores ranging from 1–2 throughout indicating that they were healthy. Final group sizes were as follows: WPC-A-EV *n* = 22, WPC-PL *n* = 19 and SL *n* = 19 (including one WPC-A-EV piglet euthanized on day 16). Body weight was similar between groups with an average daily weight gain of 34.1 ± 5.7 g/kg/day for WPC-A-EV, 33.2 ± 4.5 g/kg/day for WPC-PL and 32.9 ± 4.0 g/kg/day for SL over the 19-day study period (Appendix A). The relative organ weights did not differ amongst groups (Appendix A). The incidence of diarrhea (WPC-A-EV: 100%, WPC-PL: 89.5%, SL: 94.7%) and proportion of piglets receiving additional antibiotic treatment (WPC-A-EV: 77.3%, WPC-PL: 78.9%, SL: 68.4%) did not differ between groups.

### 3.2. In Vitro Lipolysis and Serum Triacyl-Glyceride Measurements

In vitro digestion curves depicting NaOH titration over time as a measure of free fatty acid release for each of the emulsions and corresponding intact diets are shown in Figure 1A. In the gastric phase (0–50 min), no significant differences were observed between groups with an average of 29.8 ± 6.8 µmol NaOH titrated at 50 min. During the intestinal phase (51–140 min) more digestion took place for the WPC-A-EV and WPC-PL emulsion groups relative to the SL emulsion group at all time-points except 60 min (both comparisons *p* < 0.05). In contrast, there were no differences between the intact diets at any time point in the intestinal phase. Accordingly, the total extent of digestion differed amongst emulsions (WPC-A-EV emulsion: 396 ± 2 µmol, WPC-PL emulsion: 344 ± 8 µmol, SL emulsion: 205 ± 2 µmol, all group comparisons *p* < 0.001), whereas there were no differences between intact diets (average: 819 ± 59 µmol). Higher levels of NaOH were titrated for the intact diets relative to the emulsions, likely as a result of the higher buffer capacities of the former (Appendix A). The serum TG levels at 30, 60 and 90 min after administering an oral bolus of the intact diet did not show any differences between the three groups (Figure 1B). 

### 3.3. Brain Weights, Water Content and MRI Analysis

Brain weights and water content on day 19 were similar between WPC-A-EV, WPC-PL and SL groups with an average total brain weight of 35.7 ± 2.2 g across groups (Appendix A). DTI analysis showed a lower mean diffusivity in the hippocampus in both the WPC-A-EV and WPC-PL compared to SL group (both comparisons *p* < 0.05) (Figure 2). However, no differences in mean diffusivity were observed for the remaining brain regions. Nor were there any differences for fractional anisotropy or volumetric measures in any regions (Appendix A).

### 3.4. In-Depth Lipidomics

Plasma lipidomics analysis on day 19 afforded the quantitative monitoring of 1017 individual lipid molecules and revealed a similar amount of total lipid levels in all groups (average across groups: 5718 ± 770 pmol/µL plasma). In silico-based total fatty acid analysis showed increased levels of lipid molecules with 16:1, 17:1, 19:1, 19:3, 20:1 and 20:3 fatty acyl chains in the WPC-A-EV and WPC-PL groups, whilst lipids with 18:2 fatty acyl chains were increased in the SL group. Furthermore, the total level of lipids with 20:5 fatty acyl chains were increased in the WPC-PL relative to the SL group (all *p* < 0.05) (Figure 3). At a molecular lipid species level, higher amounts of sphingomyelin 16:1;2/20:0, 32:1;2 and 38:1;2, lysoalkylphosphatidyl-choline 16:1 as well as ceramide 16:1;2/20:0 and 16:1;2/24:0, and lower levels of the sulfatide 34:2;3 were found in the WPC-A-EV and WPC-PL compared to the SL group. Additionally, the WPC-PL group had higher levels of sphingomyelin 38:1;2 relative to WPC-A-EV (all *p* < 0.05) (Appendix A). In the hippocampus, 724 individual lipid molecules were measured. There were no differences in the total amount of lipids (average across groups: 4720 ± 1454 pmol/165 µg tissue) or lipid composition of the hippocampus (Appendix A). 

### 3.5. Acquisition of Basic Motor Skills, Home Cage Activity and Short-Term Memory

Basic motor skill acquisition including time to first stand, first walk and first drink from trough were similar between WPC-A-EV, WPC-PL and SL groups (Appendix A). Home cage activity on day 2–5 was also comparable among groups (Appendix A). In the novel object recognition test, the WPC-A-EV group had a recognition index that was above and significantly different to the null preference value of 0.5 *(p* < 0.01) indicating novelty preference and recognition memory, whilst the WPC-PL (*p* = 0.05) and SL group (*p* > 0.05) did not (Figure 4A). However, no differences in recognition index or latency to explore unfamiliar objects were observed between groups (Figure 4). Fifteen piglets (four WPC-A-EV, six WPC-PL, five SL) were excluded from this analysis based on predetermined criteria, and an additional four piglets (1 WPC-A-EV, 2 WPC-PL, 1 SL) were excluded due to having had seizures. In the T-maze test, the mean percentage of correct choices over time was similar between groups (Figure 5A). On average, none of the groups reached the learning criteria of 80% by day 6. There were no differences between groups in the time it took piglets to reach the learning criteria, and seven piglets (3 WPC-A-EV, 1 WPC-PL, 3 SL) did not pass the test (Figure 5B). Piglets with seizures were also excluded from T-maze analyses along with two piglets that displayed off-task behaviour. No effects of sex were observed for any of the endpoints.

## 4. Discussion

In recent years, milk lipid research has focused on structural modifications of fat globules aiming to develop infant formulas that molecularly resemble human breast milk [20,52,53]. In line with this, the morphology of fat globules emulsified with WPC-A-EV and WPC-PL was recently characterized, and it was demonstrated that in vitro lipid hydrolysis using these dairy-derived emulsifiers was superior to SL [36]. To date, clinical and translational research has suggested roles for milk-derived polar lipids in brain development and gut physiology, maturation and immunity in neonates [24,54,55]. However, detailed knowledge of the role of EVs in neurodevelopment remains to be determined. In this study, premature piglets were fed a milk formula diet emulsified with either WPC-A-EV, WPC-PL or SL from birth to 19 days of age to determine the effects on neurodevelopment. 

The exact mechanisms by which milk-derived polar lipids may modulate neurodevelopment are still largely unknown. One proposed model is that structural modification of milk fat globules may affect intestinal lipid digestion and absorption, thus altering lipid bioavailability and brain accretion of released PUFAs or other lipid molecules—a crucial process for neurodevelopment that occurs in early life [53]. Lipid accumulation in the brain increases rapidly during the brain growth spurt [56,57], which in both humans and pigs occurs during late gestation and early postnatal life [31]. This therefore coincides with important neurodevelopmental processes such as myelination and synaptogenesis [58]. This period of accelerated brain growth is also highly sensitive to environmental stimuli and was our main rationale for using preterm piglets based on the notion that neurodevelopment could potentially be modulated by diet. Studies in rats have previously shown a connection between dietary PUFA intake, changes in brain lipid composition and behavioural outcomes suggesting an effect of dietary lipids on brain function [59,60]. 

In this study, serum TG levels following a dietary challenge bolus were similar across groups. This was supported by in vitro lipolysis data that showed that, although there were differences between WPC-A-EV, WPC-PL and SL emulsions for both intestinal and total digestion, these differences were eliminated when emulsions were combined with the remaining dietary components. This suggests that other dietary components such as proteins have emulsifying properties that override the effect of polar lipids in complex food matrices [61]. Knudsen et al. recently showed, in a similar experimental setup, that formulas containing WPC-A-EV or WPC-PL emulsifiers increased postprandial plasma TG levels compared to a formula with a SL emulsifier in age-matched preterm piglets [36]. Although the present study had a lower fat percentage in the diet and higher glucose levels in the parenteral nutrition, a follow-up study using a higher dietary fat percentage and challenge bolus did not show differences in serum TG levels between WPC-PL and SL (Appendix A). Discrepancies relative to Knudsen et al. may be due to litter variability between studies or possibly the matrix type of the collected samples (serum versus plasma) [62]. Any de novo lipogenesis related to parenteral nutrition associated glucose infusion cannot explain the absence of differences in TG levels as baseline levels were lower in this study compared to Knudsen et al.

Plasma lipidomics analysis of 19-day-old piglets confirmed similar total lipid and TG plasma levels between groups but showed specific differences, most likely reflecting the compositions of the emulsifiers, namely, that dairy phospholipids for example contain more sphingomyelin compared to soy phospholipids, whereas soy contains higher levels of lipids with 18:2 fatty acyl chains [21]. In a study comparing the plasma lipidome of infants fed a standard or MFGM-supplemented formula diet at 6 months of age, the main differences were likewise related to sphingomyelin and ceramides in addition to phosphatidyl-choline [63]. Although a causal relationship has yet to be established, these differences in the plasma lipidome were associated with improved cognition at 12 months of age [24]. Positive effects of sphingomyelin have been indicated in both translational and clinical research. For example, Oshida, et al. [27] demonstrated that dietary supplementation of sphingomyelin in developing rats, in which sphingolipid biosynthesis was inhibited, improved brain myelination compared to non-supplemented controls. Furthermore, sphingomyelin-fortified milk was positively associated with neurobehavioural development in 18-month-old preterm infants [23]. 

Although we found differences in the plasma lipid composition, this did not correlate with differences in the lipid composition of the hippocampus, which was similar between groups. This is in contrast to a study in rats that demonstrated that infant formula supplemented with MFGM reduced the differences in brain frontal lobe lipid composition (especially phosphatidylserine and phosphatidylethanolamine) between standard-formula-fed and mother-reared rats aged 18 days [64]. However, mice and rats supplemented with milk phospholipids or complex milk lipids in early life did not display differences in brain lipid composition in adulthood at 80–102 days of age relative to controls, which could indicate a transient effect early in life [53,65]. A limitation of our study is that the lipid profile was only determined in one brain region. A recent study documenting the adult mouse brain lipidome demonstrated regional differences in lipid composition [66]. It could therefore be possible that diet may prompt region-specific differences in brain lipid composition. Tracer studies of orally administered PUFAs to formula-fed neonatal piglets and baboons show that only about 0.2–4.5% reaches the cerebral cortex, although twice as much when esterified as fatty acyl chains in phospholipids compared to TGs [67,68]. Recently, it was suggested that orally administered labelled milk exosomes are able to reach the brain of mice [69]. However, this is yet to be determined for different types of milk EVs and their cargo.

MRI analysis showed no differences in brain regional volumes between groups. DTI measurements indicated microstructural changes in the hippocampus with a lower mean diffusivity for both WPC-A-EV and WPC-PL compared to SL, whilst fractional anisotropy was low but similar between groups. During brain development, fractional anisotropy and mean diffusivity both decrease in cortical grey matter reflecting dendritic arborization, synaptogenesis and cellular differentiation [70]. In deep grey matter regions, mean diffusivity also decreases during development, but fractional anisotropy has been shown to increase [70], although not specifically described for the hippocampus. It could be that mean diffusivity is a more reliable parameter than fractional anisotropy in grey matter as grey matter lacks directional diffusion [71]. Our findings may therefore suggest a higher level of hippocampal maturation in the WPC-A-EV and WPC-PL groups compared to SL. Measurements of low mean diffusivity in the hippocampus have previously been shown to correlate with better memory performance in children [72]. Moreover, mean diffusivity has been suggested to be a better predictor of hippocampal-dependent memory than volume [72]. Studies in piglets have demonstrated positive effects of dietary MFGM intake on DTI measurements in the internal capsule [28], but not in the hippocampus at 30–31 days of age [28,29]. In addition to a direct effect of milk-derived polar lipids on brain development, they may also exert indirect effects on the brain via the gut–brain axis [29]. Bioactive lipids are able to affect gut microbiota composition and in turn influence gut-brain signaling by different mechanisms including modulation of neural, immune and endocrine pathways [73,74]. Accordingly, transfaunation of preterm human infant microbiota with a low growth phenotype to pregnant gnotobiotic mice has been shown to decrease neurodevelopmental markers, increase neuroinflammation and alter neurotransmission pathways in pup brains relative to a high growth phenotype [75]. Further research investigating the involvement of the gut-brain axis in brain development secondary to dietary lipid supplementation is warranted.

In this study, there were no differences in the acquisition of basic motor skills and home cage activity levels between groups. Short-term memory, assessed by the proportion of correct choices and piglets reaching the learning criteria in a spatial T-maze test, did not differ between groups. Likewise, in a novel object recognition test, there were no differences for recognition index or latency to explore objects between groups, although only the WPC-A-EV group demonstrated novelty preference. The latter is likely to be due to large variation in the other groups, which is driven by few piglets with recognition indexes of 0. Similarly, in recent studies using the same memory tests in 17–29 day old piglets, no differences were observed between piglets fed formula diets supplemented with either MFGM [29] or MFGM in combination with prebiotics and lactoferrin [28] and those fed a non-supplemented formula diet. However, human studies disagree suggesting that there is an effect of supplementing formula with MFGM [24] or MFGM and lactoferrin [25] on cognition at 12 months of age. This could indicate that the effect of MFGM may occur later in life [29], or that the effects of MFGM are only seen after prolonged exposure. Discrepancies between human and animal studies could also be related to species–specific differences. Moreover, confounding effects such as socioeconomic status, parental intelligence and quality and quantity of stimulation of the child, that are typically eliminated in animal studies, may have a more prominent impact on cognitive development than diet [76]. 

Pigs are valuable in neurodevelopmental research as functional brain tests can be performed at a young age. In our study, the spatial T-maze test and novel object recognition test were conducted on day 12–18 and 10, respectively. Based on findings of significant sex and age effects on novel object recognition test results, Fleming and Dilger have suggested that the novel object recognition test should be conducted at least at 3 weeks of age [77]. It could therefore be that this test was conducted too early, especially in preterm piglets. On average, none of the groups in this study reached the learning criteria in the spatial T-maze test, which is in contrast to previous studies in piglets using this test [26,34]. It cannot be ruled out that palatability of the diet has influenced these results. 

Dietary interventions are likely to produce mild behavioural changes in healthy term-born infants. In this study, we attempted to enhance efficacy by using preterm piglets born at 90% of gestation. Studies comparing neurodevelopment in preterm and term piglets have demonstrated that preterm piglets in early life initially show reduced brain regional weights and delays in physical activity and some aspects of motor function relative to term-born piglets [33,34,78], but that they catch up before term-corrected age [78]. However, differences between preterm and term piglets remain in tests of balance, coordination and learning ability until at least day 18–25 [33,34]. Additionally, similar to preterm infants, preterm piglets show immaturity in some aspects of microstructural brain development at 19–25 days of age [34,79], although differences in the degree of brain maturation at birth do exist between preterm piglets and infants [79]. Ultimately, we were not able to show a dietary effect of using WPC-A-EV and WPC-PL as emulsifiers on behavioural outcomes compared to SL even under the more challenging conditions of prematurity.

## 5. Conclusions

In summary, structural modification of the surface of fat globules in infant formula by WPC-A-EV and WPC-PL emulsifiers produced changes in the composition of plasma lipids and hippocampal tissue diffusivity, but did not affect short-term memory in preterm piglets in the early postnatal period compared to SL. Under this experimental setting, our findings suggest that the differences in intestinal lipid absorption and cognition previously indicated between formula- and breastfed infants cannot fully be explained by milk-derived polar lipids. To our best knowledge, this is one of the first studies exploring the effects of EVs on neurodevelopment. Further research into the effects and potential mechanisms of action of EVs on brain development is warranted. Furthermore, it would be interesting to study the long-term and potential indirect effects of dairy-derived polar lipids on brain development. 

## Figures and Tables

**Figure 1 nutrients-13-00718-f001:**
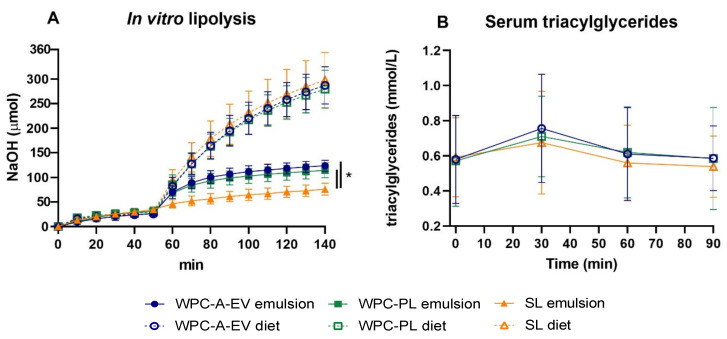
In vitro lipolysis: (**A**) NaOH titration during simulated human gastric (0–50 min) and intestinal digestion (51–140 min) of whey protein concentrate from acid whey enriched in extracellular vesicles (WPC-A-EV), whey protein concentrate enriched in phospholipids (WPC-PL) or soy lecithin (SL) emulsions (solid line, *n* = 3) and their corresponding intact diets (dashed line, *n* = 3). Serum triacylglycerides (TG): (**B**) Serum TG levels at 0, 30, 60 and 90 min after a 8 mL/kg bolus of the intact WPC-A-EV (*n* = 20–21), WPC-PL (*n* = 20–22) or SL (*n* = 20) diet. Data are presented as mean ± sd. * *p* < 0.05.

**Figure 2 nutrients-13-00718-f002:**
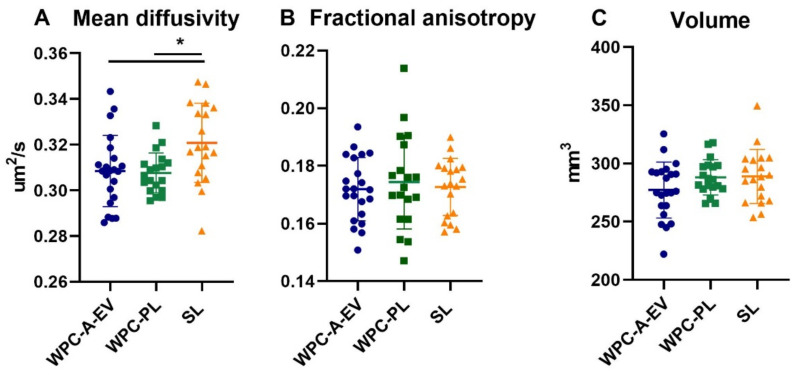
Magnetic resonance imaging (MRI) derived measurements of (**A**) mean diffusivity, (**B**) fractional anisotropy and (**C**) volume of the hippocampus in whey protein concentrate from acid whey enriched in extracellular vesicles (WPC-A-EV, blue, *n* = 22), whey protein concentrate enriched in phospholipids (WPC-PL, green, *n* = 19) and soy lecithin (SL, orange, *n* = 19) groups. Illustration of the average differences in mean diffusivity (mm^2^/s) in a coronal section of the hippocampus (outlined in red) between WPC-A-EV (**D**), SL (**E**) and WPC-PL (**F**) groups. Data are presented as mean ± sd. * *p* < 0.05.

**Figure 3 nutrients-13-00718-f003:**
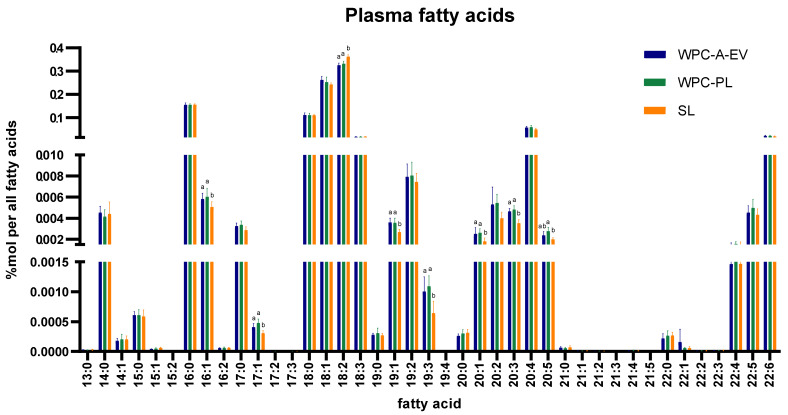
Day 19 plasma fatty acid analysis in whey protein concentrate from acid whey enriched in extracellular vesicles (WPC-A-EV, blue, *n* = 7), whey protein concentrate enriched in phospholipids (WPC-PL, green, *n* = 7) and soy lecithin (SL, orange, *n* = 7) groups. Missing values: 19:4 (1 WPC-A-EV) and 21:3 (1 WPC-A-EV, 2 WPC-PL, 2 SL). Data are presented as mean ± sd. Different lower-case letters (a and b) indicate significant (*p* < 0.05) differences between WPC-A-EV, WPC-PL and SL groups for each individual fatty acid.

**Figure 4 nutrients-13-00718-f004:**
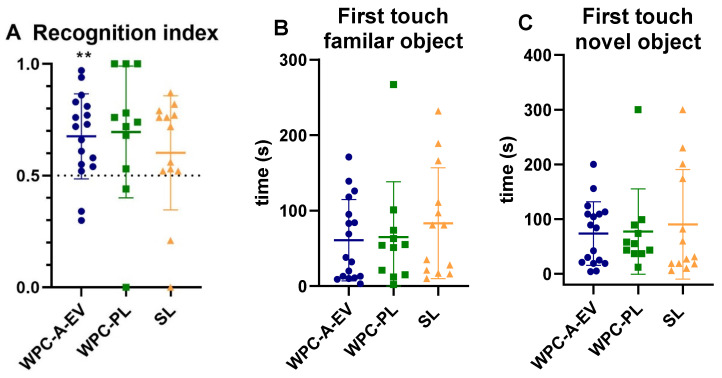
Novel object recognition test. (**A**) Recognition index for the test phase. Latency to first touch of any object in the habituation phase (**B**) and of the novel object in the test phase (**C**)**.** WPC-A-EV: whey protein concentrate from acid whey enriched in extracellular vesicles (blue, *n* = 17), WPC-PL: whey protein concentrate enriched in phospholipids (green, *n* = 11), soy lecithin: SL (orange, *n* = 13). Data are presented as mean ± sd. ** *p* < 0.01 represents the comparison between recognition index and a null preference value of 0.5.

**Figure 5 nutrients-13-00718-f005:**
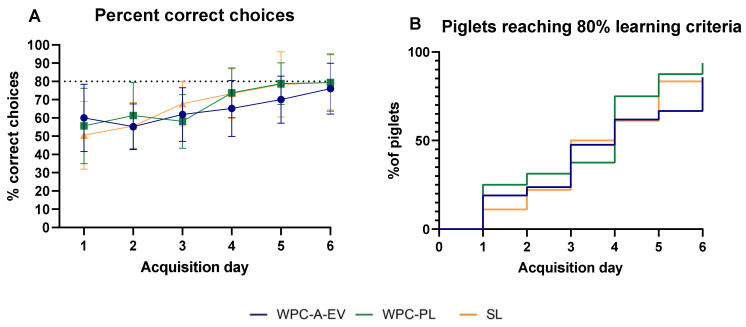
T-maze test: average percent correct choices (**A**) and percentage of piglets reaching the 80% learning criteria (**B**) for whey protein concentrate from acid whey enriched in extracellular vesicles (WPC-A-EV, blue, *n* = 20), whey protein concentrate enriched in phospholipids (WPC-PL, green, *n* = 16) and soy lecithin (SL, orange, *n* = 18) groups. Data are presented as mean ± sd.

**Table 1 nutrients-13-00718-t001:** Nutrient composition of the WPC-A-EV, WPC-PL and SL diet.

Constituents ^1^		WPC-A-EV Diet	WPC-PL Diet	SL Diet
Energy	kJ/L	3462	3460	3382
Protein	g/L	50	49	46
Carbohydrate	g/L	43	43	43
Fat	g/L	51	51	51
Docosahexaenoic acid	g/L	0.1150	0.1150	0.1150
Arachidonic acid	g/L	0.0580	0.0580	0.0580

WPC-A-EV: whey protein concentrate from acid whey enriched in extracellular vesicles; WPC-PL: whey protein concentrate enriched in phospholipids; SL: soy lecithin ^1^Lacprodan^®^ DI-9224, 25 g/L; Miprodan^®^ 40, 25 g/L; Variolac^®^ 855, 50 g/L; Phlexy-Vits, 2 g/L and emulsion, 500 g/L.

**Table 2 nutrients-13-00718-t002:** Composition of the WPC-A-EV, WPC-PL and SL emulsifiers.

Constituents (% of Sample)	WPC-A-EV	WPC-PL	SL
Dry matter	96.7	95.7	N/A
Ash	4.9	1.9	N/A
Lactose	0.23	0.5	N/A
Fat	24.1	18.0	N/A
Phospholipids	11.04	6.87	43.29
Phosphatidyl-choline	2.99	1.88	13.49
Lyso-phosphatidylcholine	0.05	0.02	0.63
Phosphatidylinositol	0.60	0.45	11.34
Lyso-phosphatidylinositol	N/A	N/A	ND
Phosphatidylserine	1.06	0.73	0.26
Lyso-phosphatidylserine	N/A	N/A	ND
Sphingomyelin	2.94	1.69	ND
Phosphatidylethanolamine	3.28	2.06	7.20
Lyso-phosphatidylethanolamine	0.06	0.02	0.25
N-acyl-phosphatidylethanolamine	N/A	N/A	3.73
Phosphatidylglycerol	N/A	N/A	0.42
Di-phosphatidylglycerol	N/A	N/A	0.44
Phosphatidic acid	0	0.01	4.34
Lysophosphatidic acid	N/A	N/A	0.16
Other phospholipids	0.07	0.01	0.96
Ganglioside D3 (mg/kg)	2942	1702	N/A
Cholesterol	1.43	0.68	N/A
Protein	65.7	73.0	N/A
Native α-lactalbumin	4.68	0.88	N/A
Native β-lactoglobulin	22.4	19.1	N/A
Native cGMP	0	2.87	N/A
Total α-lactalbumin	5.41	3.49	N/A
Total β-lactoglobulin	25.8	28.1	N/A

WPC-A-EV: whey protein concentrate from acid whey enriched in extracellular vesicles; WPC-PL: whey protein concentrate enriched in phospholipids; SL: soy lecithin; N/A: not assessed; ND: not detected.

## Data Availability

The data presented in this study are available on request from the corresponding author.

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
