# Peer review of "Dairy-Derived Emulsifiers in Infant Formula Show Marginal Effects on the Plasma Lipid Profile and Brain Structure in Preterm Piglets Relative to Soy Lecithin"

_nutrients, 2021, doi:10.3390/nu13030718_

Round 1
Reviewer 1 Report
Thank you very much for sending this excellent manuscript to Nutrients. The paper, in my opinion, is very well written and structured, with high quality introduction and discussion. The methodology, although not an expert in experimental research myself, seems quite appropriate.
I only have a couple of minor issues:
In Figure 3, it is not clear what differences are indicated by the lower-case letters “a” and “b” (between which groups). Please, clarify it.
In the results section, lines 322 – 325 and at the end of the manuscript (lines 517 - 520), a Table S5A, S5B, and S5C is mentioned, but such a Table is not provided.
Author Response
Please see attachment. A word document with responses to both reviewers

Reviewer 2 Report
This is a wonderfully clear, well-thought-out study on a physiologically relevant question.
Minor comments:
- Why was the protocol duration determined to be 19 days?
- Is the pig brain relevant to the preterm infant? Conrad/Dobbing/Odle speak to volume, what about the microcellular development and differentiation, such as myelination patterns? The preterm pig walks on its own readily after birth, so I imagine this does reflect a different maturation pattern at birth compared to humans.
- Did antibiotic usage and the presence of diarrhea differ by group? Theoretically, antibiotic exposure can be related to differences in the microbiome and there is emerging literature on the role of the microbiome in lipid metabolism.
- It would be helpful to see the distribution of deaths by group. Altered lipid hydrolysis has been shown to increase the risk of NEC. Would be of interest to see that deaths/cause of death is not different by group.
Author Response

(The authors gave the same response as above.)
